# Electrochemical and Mechanical Properties of Hexagonal Titanium Dioxide Nanotubes Formed by Sonoelectrochemical Anodization

**DOI:** 10.3390/ma17092138

**Published:** 2024-05-02

**Authors:** Katarzyna Arkusz, Aleksandra Jędrzejewska, Piotr Siwak, Mieczysław Jurczyk

**Affiliations:** 1Department of Biomedical Engineering, Faculty of Mechanical Engineering, University of Zielona Gora, 9 Licealna Street, 65-417 Zielona Gora, Poland; k.arkusz@iimb.uz.zgora.pl (K.A.); a.jedrzejewska@stud.uz.zgora.pl (A.J.); 2Institute of Mechanical Technology, Faculty of Mechanical Engineering, Poznan University of Technology, 3 Piotrowo Street, 60-965 Poznan, Poland; piotr.siwak@put.poznan.pl

**Keywords:** hexagonal titanium dioxide nanotubes (hTNTs), sonoelectrochemical anodization, morphology and composition, corrosion resistance, mechanical properties

## Abstract

This study aimed to investigate the fabrication and characterization of hexagonal titanium dioxide nanotubes (hTNTs) compared to compact TiO_2_ layers, focusing on their structural, electrochemical, corrosion, and mechanical properties. The fabrication process involved the sonoelectrochemical anodization of titanium foil in various electrolytes to obtain titanium oxide layers with different morphologies. Scanning electron microscopy revealed the formation of well-ordered hexagonal TNTs with diagonals in the range of 30–95 nm and heights in the range of 3500–4000 nm (35,000–40,000 Å). The electrochemical measurements performed in 3.5% NaCl and Ringer’s solution confirmed a more positive open-circuit potential, a lower impedance, a higher electrical conductivity, and a higher corrosion rate of hTNTs compared to the compact TiO_2_. The data revealed a major drop in the impedance modulus of hTNTs, with a diagonal of 46 ± 8 nm by 97% in 3.5% NaCl and 96% in Ringer’s solution compared to the compact TiO_2_. Nanoindentation tests revealed that the mechanical properties of the hTNTs were influenced by their diagonal size, with decreasing hardness and Young’s modulus observed with an increasing diagonal size of the hTNTs, accompanied by increased plastic deformation. Overall, these findings suggest that hTNTs exhibit promising structural and electrochemical properties, making them potential candidates for various applications, including biosensor platforms.

## 1. Introduction

Titanium (Ti) and its modified nanostructured materials are widely used in many applications, such as fuel cell technology [1], solar cells [2], biosensors [3], environmental control and photocatalytic systems [4], electronics [5], and biomaterials [6]. Ti has an excellent corrosion resistance [7,8], a good biocompatibility [9,10], thermodynamic stability [11], and a low elastic modulus [12].

Recently, titanium surface technology has evolved from bioinert surfaces such as porous titanium [13] to bioactive surfaces such as nanotextured surfaces [14]. Titanium dioxide nanotubes (TNTs) have become a focal point in titanium surface modification, offering self-organized arrays of nanoscale pores via anodization [15]. These TNTs present a distinctive structure with advantages such as increased surface areas, biocompatibility, enhanced osseointegration for bone implants [16], and potential photocatalytic applications [17]. However, challenges persist, including insufficient strength and adhesion of TNTs to the titanium substrate of implant surfaces, raising concerns about toxicity and immunoinflammatory reactions.

To address these challenges, a new class of titanium oxides with hexagonal morphologies has emerged in recent years [1,3,5]. Hexagonal titanium dioxide nanotubes (hTNTs), which are characterized by a hexagonal base and six rectangular sides, hold promise because of their unique geometry, offering potentially novel properties compared to traditional TNTs. Although circular TNTs offer ideal stress and strain distributions, the heat transfer rate is increased by introducing hexagonal tubes [18]. It has also been confirmed that the hexagonal structure of porous scaffolds can promote osteogenic differentiation and osseointegration better than a circular structure because the hexagonal pore shape has a compressive strength that matches human bone properties [19]. Thus, the mechanical, electronic, and optical properties of hTNTs are still not fully understood, making them a promising subject for investigation.

To date, research into hexagonal TiO_2_ nanotubes has taken two directions: (1) the hexagonal arrangement of circular TiO_2_ nanotubes [20,21,22,23,24], and (2) the formation of right hexagonal TNTs with a hexagonal base and six rectangular sides, referred to as hexagonal TiO_2_ nanotubes (hTNTs) [25,26,27,28,29,30,31,32,33,34,35,36,37,38].

To date, hTNTs have been formed by sonochemical-assisted chelating agent-driven anodization [28], conventional anodization/one-step anodization [32,33,34], two-step anodization [25,26,29,30,36], three-step anodization [27], pulse anodization with both positive upper and lower limits [31], anodization in a sol electrolyte [38], optimized anodization preceded by atomic layer deposition (ALD), pre-texturing by focused ion beam milling [35], and template-assisted growth using ALD [37].

The first attempt to form hexagonal TNTs was made in 2007 by Albu et al. [26] and Macak et al. [25] via a two-step high-voltage (50–60 V) anodization. However, these structures had hexagonal outer walls and circular inner walls. Similar results were obtained by other groups using the one-step anodization of Ti foil in an ethylene glycol solution containing fluoride ions [33,34,35]. Further attempts to form hTNTs included anodizing the Ti foil in an organic solution with fluoride ions in the presence of EDTA [28] and ultrasonic waves [27]. The obtained hTNTs exhibited a hexagonal shape, whereas their diagonals exceeded 100 nm. A further modification of the hTNTs preparation methods consists of two-step anodization [29,30,36], including first anodization, removing the obtained layer by mechanical peeling [36] or sonic oscillation [29,31], and second anodization. The most complicated anodizing method consists of three anodization processes, whereas the second method aims to remove the oxide layer formed during the first anodization [27]. Other anodization methods include anodization in a sol electrolyte [38] and template methods [35,37]. The formation mechanism of sunken nanotube clusters in a sol electrolyte (nitric acid solution with ethyl alcohol) involves a captivating blend of chemical and physical processes. It initiates with oxygen bubbles being generated at the nanotubes’ base; then, influenced by micelles, these bubbles rise and merge into larger ones on the tube’s surface. The pressure from these bubbles, combined with anodization, induces the creation of sunken clusters. As oxides extrude to fill the gaps between the bubbles, the characteristic hexagonal shape emerges, along with dams crucial for enhanced adhesion properties [38,39].

Among the described methods, the most homogenous and ideally hexagonal shape of hTNTs was obtained by one-step anodization [25,33,34] and template methods [35,37]. Ideal open-ended hTNTs were formed using the framework of positive template-assisted growth with ZnO nanowires and ALD [37]. However, the orientation of the hTNT and the crystallinity of the wurtzite-ZnO, anatase-TiO_2_, and cubic-Zn_2_TiO_4_ phases were not uniform, negatively affecting the mechanical and corrosion properties.

The morphology and composition of hTNTs play crucial roles in determining their performance in various applications, particularly in terms of corrosion resistance. In addition, the mechanical properties of these nanotubes are of great interest because they offer potential advantages for structural materials and other engineering applications. Understanding the relationship between the fabrication process, structural features, and functional properties of hTNTs is essential to harnessing their full potential in practical applications.

To date, there are no data in the literature related to the synthesis and characterization of hexagonal titanium dioxide nanotubes with a hexagonal base and six rectangular sides. Research on hTNTs is still in its initial stages, and a complete understanding of their synthesis, properties, and potential applications is lacking. This study aims to investigate the formation of hexagonally shaped TiO_2_ nanotubes through sonoelectrochemical anodization and pioneer elaborate electrochemical and mechanical properties depending on the morphology of the hTNTs.

## 2. Materials and Methods

### 2.1. Materials

Titanium (Ti) foil (purity 99.7%, thickness 0.25 mm), ethylene glycol (purity 99.8%), ammonium fluoride NH_4_F (purity ≥ 98%), disodium edetate Na_2_[H_2_EDTA], and sodium chloride NaCl (ACS reagent, ≥99.0%) were purchased from Sigma-Aldrich (St. Louis, MO, USA).

Ringer’s solution was prepared by dissolving one tablet (Merck, no 115525) in 500 mL neutral deionized water and then sterilizing in an autoclave (15 min at 121 °C). The final solution (500 mL) contained NaCl (1.125 g), KCl (0.0525 g), anhydrous CaCl_2_ (0.03 g), and NaHCO_3_ (0.025 g) with a pH value in the range of 6.8–7.2 at 25 °C.

All the chemical solutions were prepared using Milli-Q water.

### 2.2. Hexagonal TiO_2_ Nanotube Fabrication

The Ti foil was cut into 5 mm (width) × 20 mm (height) × 0.25 mm (thickness) pieces, sonicated in acetone and distilled water, and dried under an inert atmosphere (nitrogen). The 1-stage process was performed using a two-electrode system with a platinum sheet as the counter electrode (25 mm × 25 mm × 0.25 mm) and a titanium foil as the working electrode, with an anodized surface of 5 mm × 5 mm × 0.25 mm. The TiO_2_ layers were prepared by the electrochemical anodization of titanium foil in an electrolyte containing various concentrations of ethylene glycol solution (Table 1), 0.0818 M ammonium fluoride, and 0.003 M disodium edetate using an Autolab PGSTAT100N (Metrohm, Herisau, Switzerland) in the presence of ultrasound at a frequency of 45 kHz and a power of 200 W. Different morphologies of TiO_2_ (i.e., hexagonal and compact layers) were formed using the anodizing parameters listed in Table 1. Scanning electron microscopy (SEM) (FESEM; JEOL JSM-7600F, Tokyo, Japan) with energy-dispersive X-ray spectroscopy (EDS) was used to investigate the surface morphology and elemental composition. The surface morphologies after anodization were analyzed using field-emission scanning electron microscopy (FESEM; JEOL JSM-7600F, Tokyo, Japan) operated at 8 kV. To determine the geometric parameters of the produced layers, the outer diagonals of the nanotubes and their lengths were measured using a PCSem. The length and diagonal of the TNT were determined from the SEM micrographs at three locations for three samples. The number of images and measurements was selected to obtain 100 measurement points.

The phase compositions of the analyzed samples were determined using X-ray diffraction. This study was conducted using an X-ray Diffraction (XRD) System 3003 (GE Inspection Technologies, Alzenau, Germany) with CuKα radiation with a nickel filter, slits separated by a distance of 0.5 mm, voltage current conditions of 40 kV and 40 mA. The angular range of the 2θ scale = 20–90° was recorded using a step-counting method at a measurable step of 0.1° and a time of 3 s.

### 2.3. Electrochemical Measurements

Open-circuit potential (OCP) and electrochemical impedance spectroscopy (EIS) measurements were performed in a three-electrode configuration with compact TiO_2_ or hTNTs as the working electrode, a silver chloride electrode (E_Ag/AgCl_ = 0.222 V vs. standard hydrogen electrode), and a platinum mesh using an Autolab PGSTAT302N (Metrohm, Herisau, Switzerland). OCP tests were performed for 3600 s. The EIS spectra were recorded over the frequency range of 10^5^ to 0.1 Hz with a signal amplitude of 10 mV.

Potentiodynamic polarization curves were obtained by changing the electrode potential in the range of 250 mV around the OCP against Ag/AgCl at a scan rate of 1.0 mV/s. The corrosion potentials (E_corr_) and anodic and cathodic Tafel slopes (b_a_ and b_c_, respectively) were calculated from the polarization curves using the linear extrapolation method. The linear polarization resistance (R_p_) was determined from the slope of the current–potential plot in the range of 2 mV for the corrosion potential. The corrosion current density (i_corr_) was then calculated using the Stern–Geary equation according to Equation (1).
R_p_ = b_a_ × b_c_/(2.303(b_a_ + b_c_)i_corr_),(1)

The corrosion rate (ν_corr_) was calculated using the following equation:ν_corr_ = 3.17e^−9^ × M/(*n*FpA)i_corr_,(2)
where 3.17e^−9^ is the conversion factor from cm/s to mm/year; M (g/mol) is the atomic weight of the sample; *n* is the number of electrons exchanged in the reaction; p (g/cm^3^) is the density of the sample; F (96,485 C/mol) is the Faraday constant; and A (cm^2^) is the area of the sample.

All the experiments were performed in 3.5% NaCl and Ringer solutions.

### 2.4. Mechanical Properties’ Measurements

The mechanical properties of the compact and hexagonal TiO_2_ were measured using a Picodentor HM500 nanoindenter (Fisher, 71069 Sindelfingen, Germany). The following parameters were measured according to the ISO 14577-1 standard [40]: HM-Martens hardness, HV-Vickers hardness, EIT indentation modulus, and plastic deformation. The load was increased from zero to a maximum load of 50 mN in 20 s.

## 3. Results and Discussion

### 3.1. Microscopic and Structural Characterization of Varying Morphology of Titanium Dioxide

SEM micrographs of the surface and cross-section of compact and hexagonal TiO_2_ prepared by sonoelectrochemical anodization in NH_4_F/Na_2_[H_2_EDTA]/ethylene glycol/H_2_O electrolyte solutions according to the parameters listed in Table 1 are shown in Figure 1. The compact TiO_2_ layer (370 ± 58 nm thick) exhibits surface irregularities and no cracks. The hTNTs showed that the hexagonal nanotubes were closed at the bottom, opened from the top, and vertically oriented. hTNTs completely covered the Ti foil. The diagonal of the hTNTs increased with the increasing anodization voltage (Table 2). The SEM images showed highly ordered hexagonal TNTs with four average shorter diagonals between 30 ± 5 nm and 93 ± 13 nm and similar heights in the range of 3564–4068 nm, formed by controlling the potential from 10 V to 50 V. The hexagonal shape of the hTNTs was confirmed in a SEM picture of the hTNTs with a diagonal of 93 ± 13 nm (Figure 1c), where the bottom-view presents the hexagon and the lower part of the hTNTs presents the hexagonal outer walls and the circular inner walls, which, by reducing the thickness of the walls in the upper part, takes the form of a hexagon.

The obtained sample is hereafter referred to as diagonal X nm hTNTs.

The results of the EDS microanalysis of compact and hexagonal TiO_2_ are presented in Table 2. Despite titanium and oxide, the elemental composition of the hTNTs included fluorine residue from the sonoelectrochemical anodization process. The fluorine content increased with an increase in the hTNT diagonal length. The same phenomenon and dependency were observed in circular titanium dioxide nanotubes [41].

The formation of hTNTs involved chemical reactions associated with the formation and dissolution of the titanium dioxide layer. Initially, an ultrathin TiO_2_ passive layer was developed on the titanium surface owing to the applied potential and acidic aqueous environment, as indicated by Equation (3). Subsequently, the TiO_2_ passive layer dissolved through the action of F^−^ and [H_2_EDTA]^−2^ ions, as outlined in Equations (4)–(6) [42,43,44,45]. The applied potential drove the dissolution, the competition between the F^−^ ions and the [H_2_EDTA]^−2^ ions for complex formation with Ti^4+^, and the presence of an ultrasound, which boosted the ion mobility in the solution and guided the electrolyte to the electrode. This, in turn, accelerated the growth of hexagonal titanium dioxide nanotubes. In the final stage, the release of the F^−^ ions from the [TiF_6_]^−2^ complexes occurred through stronger EDTA ligand complexation, as shown in Equations (7) and 8).

First stage
Ti + 2H_2_O → TiO_2_ + 2H_2_↑(3)

Second stage
TiO_2_ + 6F^−^ + 4H^+^ → [TiF_6_]^2−^ + 2H_2_O(4)
TiO_2_ + [H_2_EDTA]^2−^ + 3H^+^ → [TiO(HEDTA)]^−^ + 2H_2_O(5)
TiO_2_ + [H_2_EDTA]^2−^ + 2H^+^ → [Ti(EDTA)] + 2H_2_O(6)

Third stage
[TiF_6_]^2−^ + Na_2_[H_2_EDTA] → [TiO(HEDTA)]^−^ + 12NaF(7)
[TiF_6_]^2−^ + Na_2_[H_2_EDTA] → [Ti(EDTA)] + 12NaF(8)

The X-ray diffraction analysis results presented in Figure 2 confirm the occurrence of the mainly α-Ti (substrate) phase. With the increase in the diameter of the hexagonal TiO_2_ nanotubes, the lattice constants of α-Ti decrease. The amorphous titanium dioxide in the as-prepared samples is not visible. In the subsequent step of our research, the influence of annealing on the anatase and/or rutile phase formation was studied. The mentioned phases were expected to appear after heat treatment at about 450–550 °C [46].

### 3.2. Electrochemical Characterization of Compact and Hexagonal TiO_2_ Layer

Electrochemical characterization of the compact and hexagonal TiO_2_ layers was conducted in 3.5% NaCl solutions, simulating seawater, in classical corrosion studies, as well as in Ringer’s solution, reflecting a biomedical corrosion assessment.

The samples were immersed in the electrolyte for impedance measurements until the open-circuit potential reached a steady-state value. The first measurements were performed at the open-circuit potential (OCP). Table 3 shows the average OCP values for compact TiO_2_ and hTNTs measured in both solutions at room temperature for 3600 s. A heightened potential contributes to a noble and stable surface within a given medium. This phenomenon is evident in both the hexagonal samples and the compact titanium dioxide substrate, where the anodization treatment leads to a significant surge in the OCP, also known as the corrosion potential. In the case of the Ti substrate, the standard values of OCP are −329 mV vs. SCE (−285 mV vs. Ag/AgCl), measured in 3.5% NaCl [43,44], and—400 mV vs. SCE (−356 mV vs. Ag/AgCl), measured in Ringer’s solution [45]. Anodizing resulted in significant improvements in the corrosion resistance. The compact TiO_2_ layer exhibited an OCP of −33 ± 3 mV vs. Ag/AgCl in 3.5% NaCl and 103 ± 10 mV vs. Ag/AgCl in Ringer’s solution. Sonoelectrochemical anodizing resulted in increasing the OCP in the range of −157 ± 25 mV to −213 ± 17 mV vs. Ag/AgCl in 3.5% NaCl and in the range of −138 ± 20 mV to −210 ± 31 mV vs. Ag/AgCl in Ringer’s solution. An increase in the diagonal of the hTNTs resulted in a decrease in the OCP values, indicating deterioration in the corrosion properties [46]. The higher OCP values recorded for compact TiO_2_ were the result of the absence of fluoride ions. OCP shifts to more negative values when fluoride ions are added [47]. Additionally, specimens immersed in a more acidic solution (3.5% NaCl) generally exhibit a higher corrosion potential than specimens immersed in an alkaline solution (Ringer’s) [48].

Figure 3 shows Nyquist and Bode representations of the EIS data collected for the compact and nanotubular TiO_2_ layers formed at different anodizing potentials. The data in Figure 3 show that hTNTs with different diagonals have a much higher conductivity, represented by a smaller semicircle radius in the Nyquist plots (Figure 3e,f), and an impedance modulus recorded at a low frequency (|Z| 0.1 Hz) that is an order of magnitude lower than that of compact TiO_2_, as seen from the Bode representation (Figure 3a,b), both in 3.5% NaCl and Ringer’s solution. Among the hexagonal structures, hTNTs with a diagonal of 45 ± 8 nm exhibited a much higher conductivity than the other samples. The phase angle values presented in the Bode plots (Figure 3c,d) recorded at the lowest frequency (0.1 Hz) varied from 39° to 69° in 3.5% NaCl and from 49° to 73° in Ringer’s solution, which were related to the homogeneity in the sample surface. The lowest porosity values of the phase angle (69 ± 1° and 73 ± 2° measured in 3.5% NaCl and Ringer’s solution, respectively) were observed for the compact TiO_2_ layer. The porosity of the analyzed hexagonal structures increased with the increasing diagonality of the hTNTs; that is, for the hTNTs samples with a similar height of 3500–4000 nm, a diagonal increase caused a decrease in porosity (Table 3).

The equivalent circuit [49,50,51,52] shown in Figure 3g was fitted to the EIS results for compact and hexagonal TiO_2_ (Figure 3), and the fit parameters are listed in Table 4. The components of this equivalent circuit are the electrolyte resistance (Rs), resistance (R1), and admittance (constant phase element, CPE1) of the outer tube layer (R1) and the resistance (R2) and admittance (CPE2) of the barrier layer [49]. The ohmic series’ resistance (Rs) is due to the sheet resistance corresponding to the *x*-axis value, where the first semicircle begins (on the left-hand side of Figure 3). The higher barrier-layer resistance (R1) is given by the sum of the small semicircle, which is assigned to the parallel combination of resistance and capacitance at high frequencies. The values of Rs and R1 correlate with the diagonality of hTNTs in 3.5% NaCl and Ringer’s solutions, and their values are similar to those of the compact TiO_2_ layer. The increased diagonal of the hTNTs increases the electric resistance due to the solution bulk between the electrodes (Rs) and a better corrosion resistance (R1) for 30 nm hTNTs and 45 nm hTNTs. The R2 value is given by the sum of the large semicircle at a low frequency (associated with the resistance), capacitance at the TiO_2_/electrolyte interface, and transport resistance [51]. The lower barrier-layer resistance (R2) suggests the better corrosion resistance of titanium covered with 45 nm and 80 nm hTNTs compared to compact TiO_2_ in the two tested solutions.

### 3.3. Corrosion Analysis of hTNTs

The potentiodynamic polarization curves of the compact and hexagonal TiO_2_ measured in 3.5% NaCl and Ringer’s solution are shown in Figure 4a,b. Table 5 lists the polarization parameters of the hTNTs and the compact TiO_2_ specimens. A lower i_corr_ was observed for compact TiO_2_, indicating a higher corrosion resistance of the hTNTs specimen, which is in accordance with the OCP measurements (Table 3). The corrosion current density of the hTNTs was in the range between 0.25 µA/cm^2^ and 0.38 µA/cm^2^, measured in 3.5% NaCl, and in the range of 0.12 ÷ 0.54 µA/cm^2^, measured in Ringer’s solution. In contrast, bare cp-Ti had a much higher value of approximately 30 μA/cm^2^ [53]. Similar to the OCP measurements (Table 3), the corrosion potential was higher for compact TiO_2_ (−244 mV in 3.5% NaCl and −226 mV in Ringer’s solution). hTNTs indicated a higher E_corr_ in the range of −351 ÷ −411 mV compared to the corrosion potential of the cp-Ti −0.972 V measured in 3.5% NaCl [54]. The surface modification by anodizing improved the corrosion rate from 1.88 × 10^−2^ mm/year (cp-Ti) to 2.23 × 10^−3^ mm/year (compact TiO_2_) and 1.73 × 10^−2^ mm/year in 3.5% NaCl. The obtained results confirmed that higher corrosion current values imply a weaker corrosion resistance, whereas a lower corrosion resistance is often indicated by a lower corrosion current. The i_corr_ values were directly proportional to the corrosion rate.

### 3.4. Nanoindentation Tests

The nanomechanical properties of the compact TiO_2_ and hexagonal titanium dioxide nanotubes (with diagonals in the range of 30–95 nm), such as the HM-Martens hardness, HV Vickers hardness, EIT indentation modulus, and plastic deformation, are presented in Table 6. The diagonal of the hTNTs significantly influenced the mechanical properties of the synthesized specimens. In the case of the compact TiO_2_ layer, Young’s modulus and Vickers hardness equaled 151.4 GPa and 395.16 N/mm^2^, respectively. Vickers hardness reached 154.20, 80.44, 29.75, and 41.85 N/mm^2^ for 30 nm, 45 nm, 80 nm, and 95 nm hTNTs, respectively. From the data, we can see that increasing the diagonal of the hTNTs to 80 nm decreased their hardness and Young’s modulus. However, the plastic deformation increased from 76.7% for compact TiO_2_ to 93.13% for the 80 nm hTNTs. Further increases in each mechanical property reported for the 95 nm hTNTs may be the result of exceeding the nanoscale.

Table 7 shows Young’s modulus of the circular amorphous TNT formed on the Ti foil, considering the TNT morphology. As it can be seen, the elastic modulus of the circular TNT depends on the diameter and height of the nanotubes. Additionally, the mechanical properties of the TNT depend on the electrolyte used during anodization, which has not been considered in this study. However, circular TiO_2_ nanotubes have an elastic modulus in a wide range of 4–57 GPa [55,56,57,58,59,60,61,62,63], while hexagonal TiO_2_ nanotubes have a higher elastic modulus in a range of 54–99 GPa. These results confirm that hexagonal TNTs exhibit better mechanical properties than circular TNTs.

### 3.5. Influence of Diagonality of hTNTs on Electrochemical and Mechanical Properties

The developed method for producing hTNTs enabled the formation of a nanotubular TiO_2_ layer with a uniform height of 3500–4000 nm and various diagonals of hexagons forming the hTNTs. In summary, the results presented in this study clearly indicate the influence of the diagonality of the hTNTs on the electrochemical and mechanical properties, as summarized in Table 8. Additionally, our research aimed to characterize hTNTs for various applications. Therefore, electrochemical studies were conducted in a 3.5% NaCl solution as a standard solution, allowing the evaluation of corrosion resistance for industrial applications, and in Ringer’s solution, used as a solution simulating physiological fluid, to determine the potential application of hTNTs as biomaterials.

In both 3.5% NaCl and Ringer’s solutions, the hTNTs with larger diameters exhibited a more negative open-circuit potential. The best electrical conductivity of the hTNTs was recorded for hTNTs with a diagonal of 46 ± 8 nm both in the 3.5% NaCl and Ringer’s solution. The corrosion rate and mechanical properties of the hTNTs were dependent on their diagonals in the range of 30–85 nm. The 95 nm hTNTs was close to exceeding the nanoscale, so this correlation was not observed for this sample. Increasing the diagonality of the hTNTs resulted in a decrease in the corrosion rate, a decrease in the Martens and Vickers hardness, a decrease in Young’s modulus, and an increase in plastic deformation.

The electrochemical characteristics of the hTNTs in both solutions differed. The open-circuit potential measurements showed more positive values for hTNTs in Ringer’s solution, which may indicate the deposition of Ca^2+^ or CO_3_^2−^ ions on the surface. This was further confirmed by the higher values of the impedance modulus, thereby worsening the electrical conductivity of the hTNTs in Ringer’s solution or increasing the corrosion rates. However, it should be noted that the corrosion rates measured in both Ringer’s solution and 3.5% NaCl fell within the range of the corrosion rates of materials used as biomaterials.

## 4. Conclusions

This study investigated the fabrication and characterization of hexagonal titanium dioxide nanotubes (hTNTs) compared to compact TiO_2_ layers, focusing on their structural, electrochemical, corrosion, and mechanical properties. The fabrication process involved the sonoelectrochemical anodization of titanium foil in various electrolytes to obtain TiO_2_ layers with different morphologies. Scanning electron microscopy (SEM) revealed the formation of well-ordered hexagonal TNTs with diagonals in the range of 30–95 nm and heights in the range of 3500–4000 nm. An EDS analysis confirmed the presence of fluoride in the hTNTs, and the fluoride content increased with an increase in the diagonal of the hTNTs. X-ray diffraction confirmed the amorphous phase composition of the hTNTs.

The electrochemical measurements performed in 3.5% NaCl and Ringer’s solution demonstrated the following:An increase in the diagonal of the hTNTs resulted in a decrease in the open-circuit potential values, indicating deterioration in the corrosion properties.An electrochemical impedance spectroscopy analysis revealed that the hTNTs possessed a lower impedance modulus than compact TiO_2_, indicating better charge transfer kinetics.The hTNTs exhibited lower corrosion rates and higher corrosion potentials than the compact TiO_2_ layers.

The nanoindentation tests revealed that the mechanical properties of the hTNTs were influenced by their diagonal size, with decreasing hardness and Young’s modulus observed with the increasing diagonal size of the hTNTs, accompanied by increased plastic deformation.

Overall, these findings suggest that hTNTs exhibit promising structural, electrochemical, corrosion, and mechanical properties, making them potential candidates for various applications, including as biosensor platforms.

## Figures and Tables

**Figure 1 materials-17-02138-f001:**
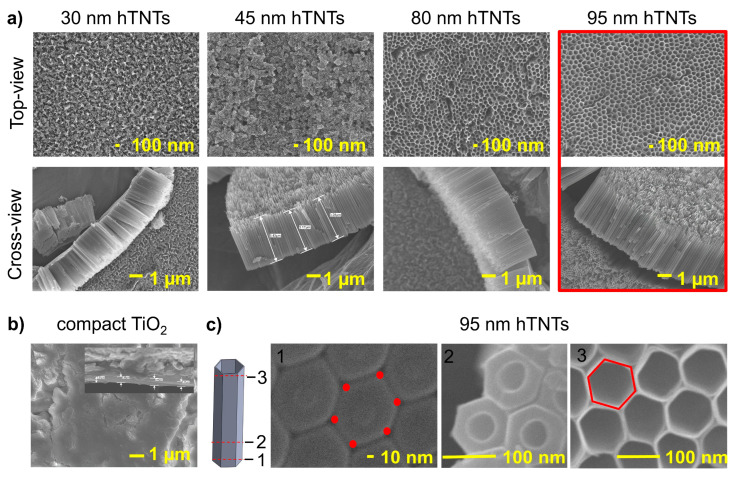
SEM top-view and cross-sectional images of hexagonal TiO_2_ nanotubes with a diagonal in the range of 30–95 nm and a height of 3500–4000 nm (**a**), compact oxide layer (**b**), and slice-sectional top-view of 95 nm hTNTs, where 1 is the bottom-view of the hTNTs, 2 is the lower part of the hTNTs, and 3 is the top surface of the hTNTs (**c**).

**Figure 2 materials-17-02138-f002:**
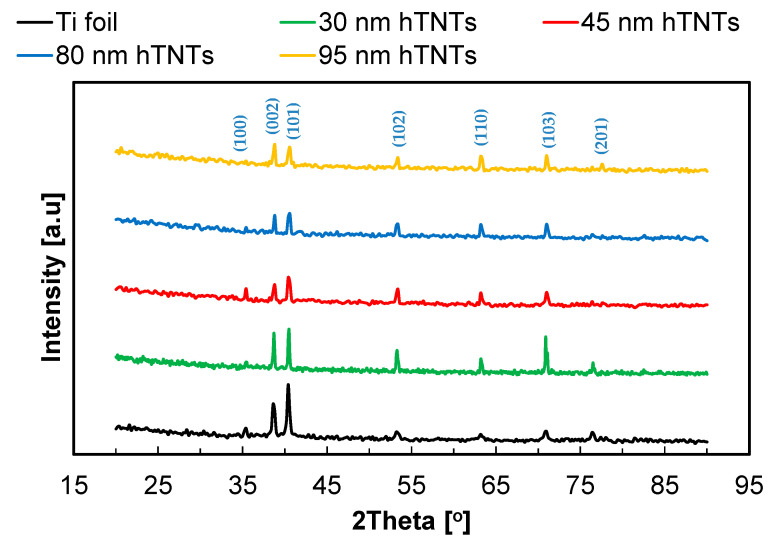
XRD spectra of hexagonal TiO_2_ nanotubes with a diagonal in the range of 30–95 nm formed on Ti foil.

**Figure 3 materials-17-02138-f003:**
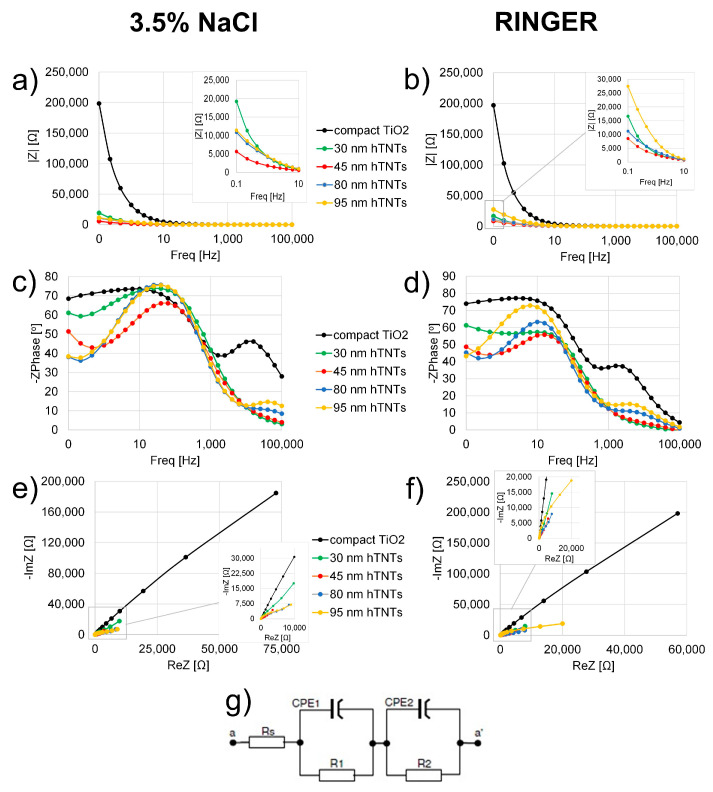
Bode (**a**–**e**) and Nyquist (**e**,**f**) plots for compact and hexagonal TiO_2_ measured in 3.5% NaCl (**a**,**c**,**e**), Ringer’s solution (**b**,**d**,**f**), and an equivalent circuit (**g**).

**Figure 4 materials-17-02138-f004:**
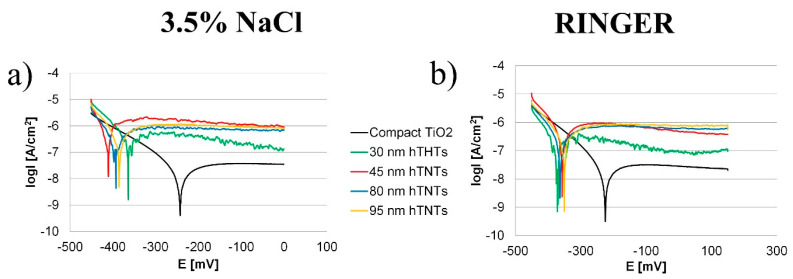
Potentiodynamic polarization curves for compact and hexagonal TiO_2_ measured in (**a**) 3.5% NaCl and (**b**) Ringer’s solution.

**Table 1 materials-17-02138-t001:** Parameters of the anodizing process for TiO_2_ layer formation with varying morphologies, such as compact layers and hexagonal nanotubular layers (hTNTs).

Type of TiO_2_ Layer	Time [min]	Potential [V]	Ethylene Glycol Concentration [%]	Ammonium Fluoride Concentration [M]	Disodium Edetate Concentration [M]
Compact	60	80	95	-	0.0030
Hexagonal	60	10	95	0.818	
	60	20	94		
	60	35	95		
	50	50	90		

**Table 2 materials-17-02138-t002:** Morphology and EDS analysis of compact (height) and hexagonal (diagonal and height) TiO_2_ formed using electrochemical anodization.

	Morphology		EDS Analysis		
Sample	Diagonal [nm]	Height [nm]	Ti [% wt.]	O [% wt.]	F [% wt.]
Compact TiO_2_	-	370 ± 58	66.88 ± 3.08	33.13 ± 3.08	-
30 nm hTNTs	30 ± 5	3707 ± 97	57.73 ± 4.40	34.26 ± 4.04	8.02 ± 0.93
45 nm hTNTs	46 ± 8	3790 ± 585	62.68 ± 2.63	28.58 ± 2.28	8.88 ± 0.39
80 nm hTNTs	82 ± 9	4068 ± 104	60.77 ± 1.85	30.05 ± 1.58	9.18 ± 0.40
95 nm hTNTs	93 ± 13	3564 ± 102	58.00 ± 3.22	31.45 ± 1.92	10.55 ± 1.38

**Table 3 materials-17-02138-t003:** Average values of impedance parameters (recorded at 0.1 Hz) and open-circuit potential of compact and hexagonal TiO_2_ samples.

Sample	|Z| [Ω]	ReZ [Ω]	−ImZ [Ω]	−Z Phase [°]	OCP [mV]
3.5% NaCl					
Compact TiO_2_	198,293 ± 34,704	71,598 ± 8582	185,054 ± 32,366	69 ± 1	−33 ± 3
30 nm hTNTs	19,097 ± 2671	9050 ± 1511	16,805 ± 2301	62 ± 2	−157 ± 25
45 nm hTNTs	5259 ± 1958	3279 ± 1210	4109 ± 1546	51 ± 2	−191 ± 17
80 nm hTNTs	10,237 ± 1033	7990 ± 823	6389 ± 749	39 ± 2	−206 ± 15
95 nm hTNTs	12,212 ± 1512	9500 ± 1436	7660 ± 698	39 ± 2	−213 ± 17
RINGER					
Compact TiO_2_	193,779 ± 17,420	56,695 ± 4348	185,230 ± 17,792	73 ± 2	103 ± 10
30 nm hTNTs	16,196 ± 1080	8018 ± 385	14,048 ± 1368	60 ± 3	−138 ± 20
45 nm hTNTs	8423 ± 1665	5549 ± 1078	6329 ± 1320	49 ± 2	−149 ± 11
80 nm hTNTs	10,960 ± 3065	7742 ± 2023	7743 ± 2362	45 ± 3	−198 ± 13
95 nm hTNTs	24,893 ± 7297	18,313 ± 5982	16,821 ± 4388	43 ± 3	−210 ± 31

**Table 4 materials-17-02138-t004:** EIS fitting results for compact and hexagonal TiO_2_ measured in 3.5% NaCl and Ringer’s solution using the equivalent circuit shown in Figure 3g.

EC Parameter	30 nm hTNTs		45 nm hTNTs		80 nm hTNTs		95 nm hTNTs		Compact TiO_2_	
	Value	SD	Value	SD	Value	SD	Value	SD	Value	SD
3.5% NaCl										
Rs [Ω∙cm^2^]	5.53	0.55	6.71	1.16	7.85	1.05	8.19	0.75	8.22	0.25
Y1 [S/cm^2^]	8.90 × 10^−5^	1.04 × 10^−5^	2.98 × 10^−4^	1.07 × 10^−4^	1.94 × 10^−4^	2.58 × 10^−5^	1.64 × 10^−4^	1.02 × 10^−5^	7.57 × 10^−6^	1.08 × 10^−6^
N1	0.81	0.03	0.69	0.01	0.71	0.01	0.71	0.01	0.83	0.01
R1 [Ω∙cm^2^]	5.60	0.59	6.86	1.23	8.06	1.12	8.65	0.91	7.10	0.25
Y2 [S/cm^2^]	1.01 × 10^−4^	3.96 × 10^−5^	1.03 × 10^−4^	4.45 × 10^−5^	3.03 × 10^−5^	3.59 × 10^−6^	3.26 × 10^−5^	8.06 × 10^−6^	2.26 × 10^−6^	3.54 × 10^−7^
N2	0.87	0.02	0.83	0.05	0.96	0.02	0.97	0.02	0.84	0.01
R2 [Ω∙cm^2^]	3729	1965	1046	507	3528	157	4147	724	67	9
χ^2^	0.04	0.01	0.04	0.01	0.24	0.06	0.38	0.07	0.05	0.01
τ1 = R1∙Y1	4.99 × 10^−4^		2.05 × 10^−3^		1.56 × 10^−3^		1.42 × 10^−3^		5.38 × 10^−5^	
τ2 = R2∙Y2	3.78 × 10^−1^		1.07 × 10^−1^		1.07 × 10^−1^		1.35 × 10^−1^		1.50 × 10^−4^	
RINGER										
	Value	SD	Value	SD	Value	SD	Value	SD	Value	SD
Rs [Ω∙cm^2^]	28.61	0.67	29.60	0.70	31.90	1.23	31.20	0.01	31.39	2.12
Y1 [S/cm^2^]	8.90 × 10^−5^	3.58 × 10^−6^	1.95 × 10^−4^	3.39 × 10^−5^	1.41 × 10^−4^	3.21 × 10^−5^	9.50 × 10^−5^	1.50 × 10^−5^	7.70 × 10^−6^	6.22 × 10^−7^
N1	0.74	0.01	0.67	0.02	0.63	0.05	0.66	0.04	0.85	0.01
R1 [Ω∙cm^2^]	46.30	2.16	49.31	2.66	58.48	5.06	61.84	5.31	50.48	6.12
Y2 [S/cm^2^]	1.38 × 10^−4^	2.55 × 10^−5^	8.01 × 10^−5^	1.52 × 10^−5^	4.02 × 10^−5^	4.02 × 10^−6^	3.77 × 10^−5^	8.42 × 10^−6^	2.26 × 10^−6^	3.20 × 10^−7^
N2	0.87	0.04	0.85	0.04	1.06	0.02	1.08	0.02	0.85	0.01
R2 [Ω∙cm^2^]	565	271	1718	522	2321	1035	4326	97	215	78
χ^2^	0.02	0.01	0.02	0.01	0.18	0.05	0.44	0.07	0.03	0.02
τ1 = R1∙Y1	4.12 × 10^−3^		9.64 × 10^−3^		8.22 × 10^−3^		5.88 × 10^−3^		3.89 × 10^−4^	
τ2 = R2∙Y2	7.81 × 10^−2^		1.38 × 10^−1^		9.34 × 10^−2^		1.63 × 10^−1^		4.86 × 10^−4^	

Rs—electrolyte resistance; Y1—admittance of the outer tube layer; N1—exponent; R1—resistance of the outer tube layer; Y2—admittance of the barrier layer; N2—exponent; R2—resistance of the barrier layer; χ^2^—Chi squared error; τ1—time constant of the outer tube layer; and τ2—time constant of the barrier layer.

**Table 5 materials-17-02138-t005:** Results of potentiodynamic polarization studies measured in 3.5% NaCl and Ringer solution, where i_corr_ is the corrosion current density, E_corr_ is the corrosion potential, and Rp is the polarization resistance.

Sample	i_corr_ [A/cm^2^]	E_corr_ [mV]	Rp [Ω/cm^2^]	Corrosion Rate [mm/year]
3.5% NaCl				
Compact TiO_2_	3.40 × 10^−8^ ± 5.29 × 10^−9^	−244 ± 2	1,381,172 ± 143,814	0.0024 ± 0.0004
30 nm hTNTs	4.01 × 10^−7^ ± 4.76 × 10^−8^	−363 ± 8	69,447 ± 9927	0.027 ± 0.0033
45 nm hTNTs	2.96 × 10^−7^ ± 9.49 × 10^−8^	−411 ± 5	20,557 ± 5518	0.021 ± 0.0034
80 nm hTNTs	2.63 × 10^−7^ ± 8.33 × 10^−9^	−393 ± 5	54,904 ± 3068	0.017 ± 0.0006
95 nm hTNTs	4.19 × 10^−7^ ± 4.08 × 10^−8^	−386 ± 1	45,702 ± 3246	0.025 ± 0.0028
RINGER				
Compact TiO_2_	1.90 × 10^−8^ ± 7.07 × 10^−10^	−226 ± 5	1,107,176 ± 262,155	0.002 ± 0.00004
30 nm hTNTs	1.22 × 10^−7^ ± 1.72 × 10^−8^	−372 ± 2	133,634 ± 7437	0.008 ± 0.001
45 nm hTNTs	5.32 × 10^−7^ ± 9.22 × 10^−8^	−357 ± 7	55,642 ± 1805	0.038 ± 0.006
80 nm hTNTs	2.76 × 10^−7^ ± 1.13 × 10^−8^	−364 ± 2	86,066 ± 15,194	0.022 ± 0.0008
95 nm hTNTs	3.41 × 10^−7^ ± 9.87 × 10^−9^	−351 ± 10	82,370 ± 6434	0.023 ± 0.0007

**Table 6 materials-17-02138-t006:** Mean values of the mechanical properties of the compact and hexagonal TiO_2_ measured by nanoindentation tests.

Sample	Martens Hardness [N/mm^2^]	Vickers Hardness [N/mm^2^]	Young’s Modulus [GPa]	Plastic Deformation [%]
Compact TiO_2_	3152.45 ± 402.33	395.16 ± 54.00	151.39 ± 19.96	76.705 ± 2.518
30 nm hTNTs	1328.48 ± 154.20	154.20 ± 32.42	99.48 ± 10.42	80.736 ± 1.372
45 nm hTNTs	711.21 ± 63.26	80.44 ± 7.07	76.49 ± 8.07	85.087 ± 0.604
80 nm hTNTs	272.93 ± 25.14	29.75 ± 2.74	54.74 ± 2.61	93.132 ± 1.067
95 nm hTNTs	379.71 ± 26.18	41.85 ± 2.99	65.19 ± 3.68	90.698 ± 0.812

**Table 7 materials-17-02138-t007:** Comparison of Young’s moduli of circular (literature data) and hexagonal (results) titanium dioxide nanotubes.

TNT (Diameter/Diagonal × Height) [nm]	Young’s Modulus [GPa]	Reference
20–150 × 210–1920	36–43	[55]
45–50 × 234–625	4.6–32.8	[56]
198 × 8500	5.1	[57]
43–58 × 234–650	36–43	[58]
31–128 × 240–3500	~10	[59]
75–110 × 7000–10,000	23−44	[60]
80 × 10,000	57	[61]
15–100 × 200	8.7–19.2	[62]
100 × 4000	~35	[63]
30–95 × 3500–4000	54–99	[this manuscript]

**Table 8 materials-17-02138-t008:** Summary of the electrochemical and mechanical properties of hTNTs with varying diameters.

Diagonal of hTNTs	OCP [mV]	|Z| [Ω]	Corrosion Rate [mm/year]	Martens Hardness [N/mm^2^]	Vickers Hardness [N/mm^2^]	Young’s Modulus [GPa]	Plastic Deformation [%]
3.5% NaCl							
30 ± 5	−157 ± 25	19,097 ± 2671	0.027 ± 0.0033	1328.48 ± 154.20	154.20 ± 32.42	99.48 ± 10.42	80.736 ± 1.372
46 ± 8	−191 ± 17	5259 ± 1958	0.021 ± 0.0034	711.21 ± 63.26	80.44 ± 7.07	76.49 ± 8.07	85.087 ± 0.604
82 ± 9	−206 ± 15	10,237 ± 1033	0.017 ± 0.0006	272.93 ± 25.14	29.75 ± 2.74	54.74 ± 2.61	93.132 ± 1.067
93 ± 13	−213 ± 17	12,212 ± 1512	0.025 ± 0.0028	379.71 ± 26.18	41.85 ± 2.99	65.19 ± 3.68	90.698 ± 0.812
RINGER							
30 ± 5	−138 ± 20	16,196 ± 1080	0.008 ± 0.001	1328.48 ± 154.20	154.20 ± 32.42	99.48 ± 10.42	80.736 ± 1.372
46 ± 8	−149 ± 11	8423 ± 1665	0.038 ± 0.006	711.21 ± 63.26	80.44 ± 7.07	76.49 ± 8.07	85.087 ± 0.604
82 ± 9	−198 ± 13	10,960 ± 3065	0.022 ± 0.0008	272.93 ± 25.14	29.75 ± 2.74	54.74 ± 2.61	93.132 ± 1.067
93 ± 13	−210 ± 31	24,893 ± 7297	0.023 ± 0.0007	379.71 ± 26.18	41.85 ± 2.99	65.19 ± 3.68	90.698 ± 0.812

## Data Availability

The data presented in this study are available upon request from the corresponding author (accurately indicating the status). Data are contained within the article.

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
