# Peer review of "Electrochemical and Mechanical Properties of Hexagonal Titanium Dioxide Nanotubes Formed by Sonoelectrochemical Anodization"

_materials, 2024, doi:10.3390/ma17092138_

Round 1

Reviewer 1 Report (Previous Reviewer 1)

Comments and Suggestions for Authors

The work has gotten significantly better. I recommend it for publication. In addition I kindly recommend the authors to note in the introduction that hedrated titanium is successfully anodized 10.3390/ma14247490, this can be used for hydrogen storage.

Author Response

Dear Reviewer

Thank you. Manuscript 10.3390/ma14247490 was cited in the Introduction. We would like to thank you once again for your suggestions for improving our manuscript.

Yours faithfully

M. Jurczyk

Reviewer 2 Report (Previous Reviewer 2)

Comments and Suggestions for Authors

Authors have addressed all my points. I can recommend the Ms for publication now.

Author Response

Dear Reviewer,

We would like to thank you once again for your suggestions for improving our manuscript.

Yours faithfully

M. Jurczyk

Reviewer 3 Report (Previous Reviewer 4)

Comments and Suggestions for Authors

Authors addressed review comments, it's suggested to be accepted.

Comments on the Quality of English Language

English is fine, minor revision will be better.

Author Response

Dear Reviewer

Thank you. We would like to thank you once again for your suggestions for improving our manuscript.

Yours faithfully,

M. Jurczyk

Reviewer 4 Report (New Reviewer)

Comments and Suggestions for Authors

Sonoelectrochemical anodization of hexagonal TiO2 nanotubes was investigated and its mechanical and electrochemical characteristics. The effort is excellent; it is thorough and methodical. I propose that the manucispt be accepted after the following little remarks are addressed.

1. Change the nm to Ao units in the abstract (3500–4000 nm into Ao). The specified range differs from the meaning of nanomaterials.

2. Authors of the introduction described the sol electrolyte method of producing TiO2 nanotubes. Would you kindly pen a few lines on it?

3. It is well known to synthesise TiO2 nanotubes by anodization in fluoride solution by etching. The novelty, according to the authors, is getting the TiO2 nanotubes with six rectangular sides and a hexagonal base. Could you perhaps clarify how the hexagonal base and six rectangular sides produce the difference from the current techniques, processes, or protocols? Beyond the chemical equations, which are already included in the book, readers would be more curious in the potential mechanism in pictorial representation.

4. The parameters listed in Table 4 should be explained. Seeing the table is difficult to get any insights because it contains a lot of information. I thus advise the authors to present and describe the trend that emerges in all conceivable EC parameters vs hTNSs size.

Author Response

see att. document

This manuscript is a resubmission of an earlier submission. The following is a list of the peer review reports and author responses from that submission.

Round 1

Reviewer 1 Report

Comments and Suggestions for Authors

Manuscript materials-2894785 Round 1

Dr. Mieczysław Jurczyk manuscript is devoted to effect of hexagonal titanium dioxide nanotubes morphology on corrosion resistance, electrochemical impedance, and mechanical properties was investigated and compared with the compact TiO2 layer. The work is experimental in nature. However, it has significant shortcomings:

1.     Honestly the work raises a lot of doubts. At first glance, it's beautiful. The authors took some new material, which was obtained in 2020, obtained as if this material already by anodizing and investigated it.

2.     Strictly speaking such hexagonally packed tubes were obtained much earlier, not in 2020. They were obtained by anodizing titanium through porous aluminum oxide. They are hexagonally packed and well researched. However, the authors do not do a review on this topic, and most importantly do not show the differences between titania obtained by anodizing through AOA pores and their titania. There are a lot of such works.

3.     Mainly the paper does not say anything about composition. If the material is so new different from tubular or columnar titanium oxide obtained a long time ago, the authors should have started with the study of composition, XPS, XRD, EDX at least. None of this is in the paper.

4.     How can a new material be investigated without information about its composition? How do the authors prove that this material is new? How does it differ from the tubular titanium obtained in these works?

5.     The keywords are repeated in the title and abstract.

6.     Line 19: OCP abbreviation not introduced

7.     Line 150: It is difficult to call the resulting arrays highly ordered. This needs to be proved.

8.     Line 121, Line 188: OCP abbreviation already introduced.

9.     Line 198–200: “mV” missed

10. 

Figure 2: Obviously, this nanotubes isn’t hexagonal. In other images, this is less visible, but the obvious presence of cylindrical tubes is visible. It is also necessary to describe the destruction of the structure, what caused it.

11.  Authors should add a description of the calculation of morphological parameters. Specify the software.

12.  Line 286: Сomparison with only once work isn’t enough for such a serious statement. It is necessary to add at least two more works. It is even better to make a comparative table

13.  The conclusion should be rewritten and begin with a general result that accomplishes the purpose of the paper noted in the introduction, followed by a paragraph-by-paragraph listing of the main results with numerical values on which the general result is based.

14.  I recommend that authors improve their figs style. Work on the layout on the sheet, bring all fonts and notations to the same form. Make the figs like Picasso's paintings, beautiful, to be admired. Also I recommend improve their table style.

15.  Are the authors anodizing? That's great. But where are the anodizing kinetics? It is not at all clear what they anodized and how. No kinetics, no compositional studies.

Sorry, unfortunately the work is of very low quality. I cannot recommend it for publication and must reject it. It is a pity that scientists with such high ranking and positions try to promote manuscripts of such quality.

Comments on the Quality of English Language

A large number of introductory words and conjunctions. The sentences are written in Polish, not English. It is recommended to work on the style.

Reviewer 2 Report

Comments and Suggestions for Authors

This Ms on the first glance appears interesting, but upon a thorough look, there are many problems. The Ms is scientifically misleading in this stage and not publishable. Major problems are these

1) Authors claim to present a detailed investigation of the electrochemical and mechanical properties of hexagonal titanium dioxide nanotubes formed by sonoelectrochemical anodization. However, In contrast to literature reporting synthesis of hexagonally shaped and packed TNTs, which is dully cited, authors fail to show that such hexagonall features are present within their TNT layers. There is not a single SEM image that would demonstrates that, again, in contrast to dully cited papers.  But SEM evidence is crucial for that, especially demonstration of tube bottom parts. Authors need to provide evidence for that. As a skilled anodizer, I can already see from the top view images, that the TNTs are not very hexagonally packed and shaped.

2) Authors further claim that diagonal of hTNTs strongly influences their corrosion, impedance, and mechanical properties. However, none of this is demonstrated and convincingly presented. Authors do not introduce any measures for how to actually derive the diagonality of the TNT layers and do not present the values of diagonality in a reasonable way, such as e.g. via Table. In fact, there are many tables, whose added values is low to zero, but this table would be really welcome. Since this part is absent, it is also not clear, how does the diagonal of the hTNTs influence the mechanical properties? Authors need to improve all the introduction and discussion about diagonality

3) authors compare performance of their (claimed to be “hexagonal”) TNTs with other nanotubes, but the problem is that they used data on published “other” TNTs, not theirs. So such comparison is not very convincing and fair. One need to compare apples and apples…other might have different dimensions of TNTs, and most likely also they do have that..

Minor, yet important points:

11)      Introduction: first sentence, first word: “Titanium”, but authors mix up Ti and TiO2 here, at least, the papers that they cite are mostly on TiO2, not on Ti.

22)      Authors use in the title “sono”, again, this is misleading, there is not a word on that in the manuscript, in particular in the Experimental part.

33)      Amount of salts in solution should be expressed in molar concentrations, not wt. %.

44)      Results and discussion: EDX can be completely omitted, they do not contribute to anything and have been shown in many papers before.

55)      In Section 3. Results and discussion, the authors mention “fluorine content increases with the increase of the hTNT diagonal to more excellent wettability” without any References or any water contact angle results in the Ms.

66)      Although the authors mentions in the abstract as “These nanotubes exhibit potential applications across diverse fields, underscoring the importance of further modifications for enhanced utility ”, it is important to mention the specific potential application of the hTNTs synthesized in this study.

77)      Authors should mention the differences in electrochemical properties of compact and hexagonal TiO2 measured in 3.5% NaCl and Ringer solution observed.

Comments on the Quality of English Language

English is good enough

Reviewer 3 Report

Comments and Suggestions for Authors

Review of the Article "Materials"

The article describes the synthesis and characterization of electrochemically grown hexagonal TiO2 coatings. However, the review identifies several shortcomings that must be addressed for future consideration. The literature foundation is inadequate, as various presented information lacks references. Additionally, the authors make unsupported assumptions based on experimental results.

Regarding experimental data, while present, there is a need for a more detailed analysis, including standard deviation and statistical analyses, to differentiate between the investigated conditions. Consequently, I believe the article should not be accepted for publication. Nevertheless, I highlight some points for improvement in future submissions.

Abstract

  1. 1 - The term "nanotube diagonal" is unclear. Is it referring to diameter or length? It is recommended that it be corrected with the appropriate term.
  2. 2 - The phrase "improved the electrochemical impedance" is vague. Specify which parameters related to the impedance technique were enhanced.

Introduction

  1. 1 - When introducing the theme "Addressing these challenges, a new class of titanium oxides with a hexagonal morphology has emerged in recent years." (Line 47, page 2), it is advisable to cite similar studies to support the discussion.
  2. 2 - Caution when citing published works, such as reference 18. Simulations in macroscopic systems should not be generalized to nanoscale systems.
  3. 3 - Reference 19 mentions "3D printed hollow, hexagonal, titanium alloy porous scaffold." Justifying the superiority of this morphology requires comparative data.
  4. 4 Avoid statements not supported by references in the literature regarding TiO2-hexagonal.
  5. 5 - A thorough review of the cited references is essential to ensure consistency and similarity with previous publications before accepting the manuscript.

Methods and Results

  1. 6 - Clarify the proportion of water in the synthesis solution, as the emphasis should be on the predominant solvent, ethylene glycol.
  2. 7 - It does not make sense for the authors to have synthesized films applying 80 V using a a high current potentiostat/galvanostat, with a compliance voltage of 30 V.
  3. Results and Discussion
  4. 8 - Provide clearer images to prove the hexagonal geometry of the nanotubes. Current images show non-uniform morphology.
  5. 9 - EDS analysis is insufficient to claim the formation of non-stoichiometric TiO2 in coatings.
  6. 10 - Describing a multi-step mechanism for forming hexagonal nanotubes without references for support is inadequate.
  1. 11 - Corrosion studies should include statistical analyses like ANOVA and Tukey's test to ensure reproducibility and avoid erroneous conclusions.
  2. 12 - Include the standard deviation associated with each analysis in the presented curves for greater data robustness.
  3. 13 - Potentiodynamic polarization curves and Tafel results require replicates, standard deviation, and statistical analysis for reliable conclusions.
  4. 14 - The article needs grammatical and linguistic revision before resubmission.

I hope these suggestions enhance your article.

Comments on the Quality of English Language

The quality of the English Language can be improved.

Reviewer 4 Report

Comments and Suggestions for Authors

1, The language should be improved. For example, in the abstract, “from 30-95 nm” should be from 30 nm to 95 nm” or “in the range of 30-95 nm”; “from an anodization” is better written as “from the anodization”. There are some tense inconsistencies for the discussion of the experiment.

2, The hTNTs from the SEM images don’t show clear hexagonal structures, but more like circular shapes. The heights of hTNTs according to Table 2 don’t match the structures illustrated in Fig. 1.

The article should be revised for re-consideration.

Comments on the Quality of English Language

The language should be improved. For example, in the abstract, “from 30-95 nm” should be from 30 nm to 95 nm” or “in the range of 30-95 nm”; “from an anodization” is better written as “from the anodization”. There are some tense inconsistencies for the discussion of the experiment.